# Longer Exposure to Left-to-Right Shunts Is a Risk Factor for Pulmonary Vein Stenosis in Patients with Trisomy 21

**DOI:** 10.3390/children8010019

**Published:** 2021-01-01

**Authors:** Connie Choi, Kimberlee Gauvreau, Philip Levy, Ryan Callahan, Kathy J. Jenkins, Minghui Chen

**Affiliations:** 1Department of Cardiology, Harvard Medical School, Boston Children’s Hospital, Boston, MA 02115, USA; Kimberlee.Gauvreau@cardio.chboston.org (K.G.); ryan.callahan@cardio.chboston.org (R.C.); kathy.jenkins@cardio.chboston.org (K.J.J.); Minghui.chen@cardio.chboston.org (M.C.); 2Division of Newborn Medicine, Department of Pediatrics, Harvard Medical School, Boston Children’s Hospital, Boston, MA 02115, USA; philip.levy@childrens.harvard.edu; 3Division of Genetics and Genomics, Department of Pediatrics, Harvard Medical School, Boston Children’s Hospital, Boston, MA 02115, USA

**Keywords:** down syndrome, Trisomy 21, pulmonary vein stenosis, prematurity, congenital heart disease

## Abstract

We conducted a study to determine whether patients born with Trisomy 21 and left-to-right shunts who develop pulmonary vein stenosis (PVS) have a longer exposure to shunt physiology compared to those who do not develop PVS. We included patients seen at Boston Children’s Hospital between 15 August 2006 and 31 August 2017 born with Trisomy 21 and left-to-right shunts who developed PVS within 24 months of age. We conducted a retrospective 3:1 matched case–control study. The primary predictor was length of exposure to shunt as defined as date of birth to the first echocardiogram showing mild or no shunt. Case patients with PVS were more likely to have a longer exposure to shunt than patients in the control group (6 vs. 3 months, *p*-value 0.002). Additionally, PVS patients were also more likely to have their initial repair ≥ 4 months of age (81% vs. 42%, *p*-value 0.003) and have a gestational age ≤ 35 weeks (48% vs. 13%, *p*-value 0.003). Time exposed to shunts may be an important modifiable risk factor for PVS in patients with Trisomy 21.

## 1. Introduction

Pulmonary vein stenosis (PVS) is a rare disease in pediatric patients that has a high rate of morbidity and mortality despite medical, catheterization and surgical interventions [1,2]. This is particularly true of patients with involvement of multiple vessels [3,4]. There are currently no known methods to prevent PVS. 

The pathophysiology of PVS is unknown, but both genetic and clinical factors are likely to contribute to PVS [5]. PVS has been identified in patients with genetic syndromes such as Smith–Lemli–Opitz syndrome and Trisomy 21 [6,7]. However, a specific gene or molecular pathway has not been identified. Myofibroblast-like cells have been implicated in PVS, but the underlying causal factors are unknown [8].

The three clinical factors that have been associated with PVS are prematurity, pulmonary hypertension and total anomalous pulmonary venous return (TAPVR) [9,10]. Although the mechanism is not known, it has been speculated that abnormal pulmonary vasculature and possible states of chronic inflammation such as bronchopulmonary dysplasia may play a role in the development of PVS. Both premature infants and those with Trisomy 21 are at higher risk for pulmonary hypertension. The literature suggests that PVS may be underdiagnosed in these populations as a cause of pulmonary hypertension [11]. 

Many patients with PVS also have other associated congenital heart disease (CHD), and prior studies suggest that there are intrinsic factors including baseline anatomy that may increase the risk of post-operative PVS [12]. Anecdotally, we have noticed that patients who have anatomical variants that may cause abnormal flow in pulmonary veins are more likely to develop PVS. CHD lesions with left-to-right shunts, that by definition increase the flow through the pulmonary bed, are one such example. However, the possible association of exposure to shunts and PVS has not yet been studied. 

The primary objective of this study is to determine whether longer exposure to shunt physiology is associated with PVS. To best control for confounders, we selected a genetically homogeneous group of patients as well as compared patients with similar anatomical defects. Our cohort consisted of patients born with Trisomy 21 and left-to-right shunts. We hypothesized that patients who had PVS were more likely to have longer exposure to left-to-right shunts. Additionally, we predicted that the patients with PVS were older at the time of primary repair of the shunt and were more likely to be premature.

## 2. Materials and Methods

### 2.1. Patient Selection

We included all patients (*n* = 23) seen at Boston Children’s Hospital between 15 August 2006 and 31 August 2017 born with Trisomy 21 and left-to-right shunts who developed PVS within 24 months of age. We conducted a retrospective 3:1 matched case–control study. Each of the case patients was matched to three control patients who had Trisomy 21, same primary anatomical diagnosis, date of birth within two years of that of the control patient and no PVS (*n* = 69). PVS was defined as having at least one pulmonary vein on echocardiogram or catheterization with a gradient ≥ 4 mmHg.

### 2.2. Primary and Secondary Predictors

The primary predictor was time of exposure to left-to-right shunt defined as the date of birth to the first echocardiogram showing no significant shunt (significant shunt was defined as at least a moderate-sized defect by the echocardiogram report). If moderate or severe shunt was present beyond 24 months of age, the time of shunt exposure was defined as 24 months as this was the maximum follow-up time. The secondary predictors included (1) age of initial repair of the primary cardiac defect < 4 months versus ≥ 4 months or no repair, (2) gestational age ≤ 35 weeks or >35 weeks (patients without a numerical gestational age, but documented as full term were designated as 38 weeks). Conditional logistical regression was used for univariate and multivariable analyses. 

### 2.3. Subanalyses

Analyses were also performed on a subset of PVS patients with multivessel disease (*n* = 17) including those with PVS found in 2 or more pulmonary veins and their matched controls (*n* = 51). Additional analyses were conducted on the subset of PVS patients with a gestational age > 35 weeks (*n* = 12) with their matched controls (*n* = 36).

### 2.4. Ethics Statement

This study was approved by the Boston Children’s Hospital Institutional Review Board (protocol number: IRB-P00030684). All research was performed in accordance with relevant guidelines and regulations. The requirement for written consent was waived by the Institutional Review Board. 

## 3. Results

### 3.1. Patient Characteristics

Table 1 outlines the demographic data of the patients included in this study. There were notable differences in the surgical interventions received by the two groups. A total of 100% of the control patients and 91% of the PVS patients had their CHD repaired by 24 months. None of the control patients had a pulmonary artery (PA) band as an initial palliation prior to definitive repair, but two patients with PVS initially had PA bands. The two patients who had PA bands both had common atrioventricular canal defects and had their canal repair before 24 months of age. 

The PVS patients had a high rate of mortality and morbidity. A total of 17% of PVS patients and 0% of control patients died within the 24 months of age. PVS patients underwent 0–2 pulmonary vein surgeries and 0–15 cardiac catheterizations. The median age at diagnosis of PVS was 5.3 months, and 74% of them had multivessel disease. A total of 60% of patients with PVS had a clinical diagnosis of pulmonary hypertension, and all of these patients were treated either with Sildenafil or oxygen. Nine of the fourteen patients with a diagnosis of pulmonary hypertension had catheterizations that confirmed a mean pulmonary arterial pressure greater than 25 mmHg.

### 3.2. Primary and Secondary Predictors

Table 2 shows the univariate and multivariable analyses of the primary and secondary predictors. In the univariate analysis, PVS patients were more likely to have a longer exposure to shunt than patients in the control group (*p*-value 0.002). The median months of exposure to shunt in the PVS group was 6, compared to the control group who had a median exposure time of 3 months. Additionally, PVS patients were also more likely to have their CHD repair ≥ 4 months of age (81% vs. 42%, *p*-value 0.003) and have a gestational age ≤ 35 weeks (48% vs. 13%, *p*-value 0.003). These findings were consistent in the multivariable analysis. 

### 3.3. Multivessel PVS

Table 3 shows the subanalysis performed on PVS patients with multivessel disease (*n* = 17) and their matched controls (*n* = 51). The types of baseline anatomy in the multivessel PVS group were similar to that of the overall PVS group. The median time of diagnosis of PVS was also similar at 5.4 months. The two patients who did not have their CHD repaired were both in the multivessel PVS group as were the two patients who were initially palliated with a PA band. All four patients who died within 24 months had multivessel disease. Of note, thirteen out of the fourteen patients with PVS and pulmonary hypertension had multivessel PVS. 

Table 4 demonstrates the univariate and multivariable analyses of the primary and secondary predictors. The univariate analysis showed that patients with multivessel PVS had a longer time of exposure to shunt (*p*-value 0.006). The median months of exposure to shunt in the multivessel PVS group was 6, compared to the control group who had a median exposure time of 3 months. Again, PVS patients were also more likely to have their CHD repair ≥ 4 months of age (73% vs. 47%, *p*-value 0.045) and have a gestational age ≤ 35 weeks (53% vs. 14%, *p*-value 0.005). The multivariable analysis demonstrated the same findings. 

### 3.4. Subanalysis of Patients Based on Gestational Age

An additional subanalysis (Table 5) was performed on PVS patients with gestational age ≤ 35 weeks (*n* = 11) and their controls (*n* = 33), as well as PVS patients with gestational age > 35 weeks (*n* = 12) and their controls (*n* = 36). Table 5 shows that the time of exposure to shunt was again found to be significantly longer in those with PVS in both subgroups (≤35 weeks: *p*-value 0.023; >35 weeks: *p*-value 0.028).

## 4. Discussion

Our study demonstrated that Trisomy 21 patients with left-to-right shunts and PVS have a longer exposure to shunt than Trisomy 21 patients with left-to-right shunts and no PVS. In addition, patients who did not develop PVS were more likely to have their primary cardiac repair before the age of four months. Patients with multivessel PVS were also more likely to have longer exposure to shunts compared to controls. This is particularly important because multivessel PVS is associated with high rates of morbidity and mortality. Interestingly, the two patients who did not have their primary repair by 24 months both had multivessel disease.

Our data corroborated findings in a previously published meta-analysis on PVS [4]. Both studies showed that the median age of diagnosis of PVS was around 5 months. In addition, our results were consistent with previously published literature showing that prematurity was associated with PVS [10]. Trisomy 21 is associated with a lower gestational age at birth [13]. Our study showed that prematurity was an independent risk factor for PVS amongst patients with Trisomy 21. On the other hand, length of exposure to shunt was also significantly longer in patients who developed PVS and had a gestational age > 35 weeks. This suggests that longer exposure to shunt may be associated with PVS independent of prematurity.

There are currently no known methods for prevention of PVS. Timing of the initial repair of left-to-right shunts and repair of any significant residual shunts may be modifiable risk factors for PVS in patients with Trisomy 21 and left-to-right shunts. There is currently a wide range of ages at which the primary repair is performed for isolated CAVC. Trisomy 21 does not appear to be an independent risk factor for surgical outcome [14]. Age at repair has also not been found to increase the risk of residual VSDs, left-sided atrioventricular valve dysfunction or surgical reintervention [15,16]. Therefore, earlier surgical repair (<4 months of age) may be safe in some patients with Trisomy 21 and CAVC, and potentially decrease the likelihood of PVS in these patients.

The findings of this study suggest that physicians who are taking care of Trisomy 21 patients, especially premature Trisomy 21 patients, should be vigilant in screening for PVS. Furthermore, Trisomy 21 patients with pulmonary hypertension should be carefully assessed for PVS since this may be an important clinical manifestation of PVS. In addition, there are few publications to inform clinicians on how to minimize the development of PVS in patients with CHD, with the exception of TAPVR. Clinicians may consider early repair of all significant left-to-right shunts in Trisomy 21 patients as a potential strategy to reduce PVS occurrence. However, risk for PVS is only one component in determining timing of surgery, and as baseline risk is low, decisions should be individualized. 

One possible explanation for the association of the development of PVS and time of exposure of shunt may be due to abnormal flow states and shear stress on vascular walls caused by shunt physiology. Veins have been known to remodel and form neotima in response to external stimuli [17]. Premature infants who are more likely to have immature or abnormal pulmonary angiogenesis may be more sensitive to flow states associated with shunts [18]. However, further investigations must be performed to determine the validity of these hypotheses and the mechanism of disease. 

One limitation of this study is that due to the low prevalence of PVS, a cohort or prospective study design was not feasible. In addition, we chose a specific population with the same genetic syndrome and similar anatomical defects in order to minimize confounders. However, this means that our findings may not be generalizable to a wider range of patients who do not have Trisomy 21 and left-to-right shunts. There are also limitations of applicability as this was a single-institution study. There was a significant difference in the PVS and control groups in their rate of premature patients, but due to the small number of overall patients, there was a significant range in the odds ratio. The follow-up time was also limited to 24 months, so long-term outcomes were not assessed in this study. This may partly explain the lower incidence of pulmonary hypertension in our study (17% of case and control patients) compared to prior studies that showed 27–34% of Down syndrome patients have a diagnosis of pulmonary hypertension [19,20].

Since this study was conducted at a tertiary referral center, there may have been a selection bias towards more severely affected patients. Several patients who had PVS and residual shunt lesions were referred from outside institutions, whereas almost all patients in the control group received their care primarily from Boston Children’s Hospital. However, due to the rarity of PVS patients, this could not be avoided. Finally, it was not possible to quantify the amount of left-to-right shunt because patients did not routinely have cardiac MRIs or undergo cardiac catheterization. Therefore, we used echocardiogram findings to determine the severity of shunt.

Additional studies with a more heterogeneous group of patients and larger sample size in the future will help determine whether the length of exposure to shunt is associated with PVS in other patient populations. This will be important for clinical decision making on timing of interventions. Further investigations to explore if the degree of shunt correlates with the severity of PVS may strengthen the hypothesis that shunts play a role in the development of PVS. Finally, future efforts should be made to determine the mechanism of PVS, which will be critical in the design of effective medical and surgical interventions. 

In conclusion, patients with Trisomy 21 and left-to-right shunts who develop PVS have a longer exposure to shunts than those who have Trisomy 21, left-to-right shunts and no PVS. This is the first study to show that a modifiable risk factor such as length of exposure to shunt may be associated with PVS.

## Figures and Tables

**Table 1 children-08-00019-t001:** Patient Demographics.

Characteristics	PVS (*n* = 23)	No PVS (*n* = 69)	*p*-Value
Gestational age at birth	38 (34, 38)	38 (37, 38)	<0.01
Sex (female)	11 (49%)	33 (49%)	0.91
Anatomy			
CAVC	18 (78%)	54 (78%)	N/A
Secundum ASD	2 (9%)	5 (7%)	N/A
Membranous VSD	2 (9%)	9 (13%)	N/A
PDA	1 (4%)	1 (1%)	N/A
Surgical history			
Primary repair	21 (91%)	69 (100%)	0.01
Palliation with PA band	2 (9%)	0	0.01
Age at primary repair (months)	5 (4, 7)	3 (2, 6)	0.08
Death	4 (17%)	0	<0.01
Pulmonary vein disease			
Age at diagnosis (months)	5.3 (3.9, 8.6)	N/A	N/A
Single vessel	6 (26%)	N/A	N/A
Multivessel	17 (74%)	N/A	N/A
Number of surgeries	1 (0, 2)	N/A	N/A
Number of catheterizations	1 (0, 15)	N/A	N/A
Pulmonary hypertension			
Clinical diagnosis	14 (60%)	1 (1.4%)	<0.01
Mean PA pressure > 25 mmHg	9 (39%)	0	<0.01
Sildenafil or oxygen	14 (60%)	0	<0.01

All data is presented as numbers (percentages) or median (interquartile range), except for the number of surgery and catheter interventions, which includes the total range in parenthesis. CAVC: complete atrioventricular canal; ASD: atrial septal defect; VSD: ventricular septal defect, PDA: patent ductus arteriosus; PA: pulmonary artery.

**Table 2 children-08-00019-t002:** Univariate and Multivariate Analyses of Primary and Secondary Predictors.

Analysis	PVS (*n* = 23)	No PVS (*n* = 69)	*p*-Value
Univariate			
Time of exposure to shunt (months)	6 (4, 16)	3 (2, 6)	0.002
Age at primary repair (≥4 months)	17 (81%)	29 (42%)	0.003
Premature (birth ≤ 35 weeks GA)	11 (48%)	9 (13%)	0.003
Multivariate	Odds Ratio	95% CI	
Risk of PVS per month of exposure to shunt	1.21	(1.06, 1.39)	0.007
Premature (birth ≤ 35 weeks GA)	4.77	(1.36, 16.8)	0.015

All data is presented at numbers (percentages) or median (interquartile range) for the univariate analysis and odds ratio with 95% confidence intervals for the multivariate analysis. GA: gestational age, CI: confidence interval.

**Table 3 children-08-00019-t003:** Characteristics of Patients with Multivessel Pulmonary Vein Stenosis.

Characteristics	PVS (*n* = 17)	No PVS (*n* = 51)	*p*-Value
Gestational age at birth	35 (34, 38)	38 (37, 38)	<0.01
Sex (female)	10 (59%)	24 (47%)	0.41
Anatomy			
CAVC	12 (71%)	36 (70%)	N/A
Secundum ASD	3 (18%)	10 (20%)	N/A
Membranous VSD	2 (12%)	5 (10%)	N/A
Surgical history			
Primary repair	15 (88%)	51 (100%)	0.01
Palliation with PA band	2 (12%)	0	0.01
Age at primary repair	5 (3.5, 8.5)	3 (2, 7)	0.07
Death	4 (24%)	0	<0.01
Pulmonary vein disease			
Age at diagnosis (months)	5.4 (3.9, 8.6)	N/A	N/A
Number of surgeries	1 (0, 2)	N/A	N/A
Number of catheterizations	1 (0, 15)	N/A	N/A
Pulmonary hypertension			
Clinical diagnosis	13 (76%)	1 (2%)	<0.01
Mean PA pressure > 25 mmHg	8 (74%)	0	<0.01
Sildenafil or oxygen	13 (76%)	0	<0.01

All data is presented at numbers (percentages) or median (interquartile range), except for the number of surgery and catheter interventions, which includes the total range in parenthesis.

**Table 4 children-08-00019-t004:** Subanalysis of Patients with Multivessel Pulmonary Vein Stenosis.

Analysis	PVS (*n* = 17)	No PVS (*n* = 51)	*p*-Value
Univariate			
Time of exposure to shunt (months)	6 (4, 17)	3 (2, 7)	0.006
Age at primary repair (≥4 months)	11 (73%)	24 (47%)	0.045
Premature (birth ≤ 35 weeks GA)	9 (53%)	7 (14%)	0.005
Multivariate	Odds Ratio	95% CI	
Risk of PVS per month of exposure to shunt	1.15	(1.01, 1.30)	0.032
Premature (birth ≤ 35 weeks GA)	4.33	(1.14, 16.4)	0.031

All data is presented at numbers (percentages) or median (interquartile range) for the univariate analysis and odds ratio with 95% confidence intervals for the multivariate analysis.

**Table 5 children-08-00019-t005:** Time of Exposure to Shunt for Patients Based on Gestational Age.

Analysis	PVS	No PVS	*p*-Value
Gestational age at birth ≤ 35 weeks			
Time of exposure to shunt (months)	6 (4, 24)	3 (2, 5)	0.023
Gestational age at birth > 35 weeks			
Time of exposure to shunt (months)	8 (4, 10)	3 (2, 6)	0.028

All data is presented as the median (interquartile range).

## Data Availability

The data presented in this study are available on request from the corresponding author. The data are not publicly available in order to maintain patient privacy.

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
