# Peer review of "Longer Exposure to Left-to-Right Shunts Is a Risk Factor for Pulmonary Vein Stenosis in Patients with Trisomy 21"

_children, 2021, doi:10.3390/children8010019_

Round 1

Reviewer 1 Report

doct Choi and coauthors reported an interesting study on the possible role of Left to right shunts on pulmonary vein stenosis in down syndrome. 

this paper is interesting, background is extensive and references are clear and updated. methods are well conducted and the results are in line, clear and well reported in adeguate tables. 

they also data regarding two subgroups with single or multivessels involved in PVS

Just few comments:

  •  in both results (table 2 and table 4) CI for premature show an interval very large... how do they explain these results? may be in the discussion, the authors should consider this data and speculate the reasons for these statistical results.
  • in table 1 and table, I would add the age at surgery for both groups. 

Author Response

Point 1: The confidence interval for the odds ratio for prematurity between the PVS and no PVS group appears to be large in both the initial analysis as well as in the multivessel PVS subgroup analysis.

Response 1: The variable was binary (ie. gestational age ≤ 35 weeks vs gestational age > 35 weeks) and the sample size of the patients was small, which led to significant variability. However, the analysis still showed a statistically significant difference. We added this comment to the discussion.

Point 2: In table 1 and 3, I would add the age at surgery of both groups.

Response 2: We added these values to table 1 and 3.

Reviewer 2 Report

The authors have provided a clearly written manuscript that adequately supports the relationship between duration of exposure to shunt physiology and the development of pulmonary vein stenosis. One of the most interesting aspects of this manuscript is that pulmonary vein stenosis may develop independent of prematurity in patients with Trisomy 21. Is there any way for the authors to provide p-values for table 1 to denote significant differences between the PVS versus non-PVS groups? I was also curious to learn more details of those patients who were palliated with a PA band. Did this palliation seem to make any difference to their clinical course? There a couple of grammatical errors.

Author Response

Point 1: Is there any way for the authors to provide p-values for table 1 to denote significant differences between the PVS versus non-PVS groups?

Response 1: The p-valves were added to the appropriate rows.

Point 2:  I was also curious to learn more details of those patients who were palliated with a PA band. Did this palliation seem to make any difference to their clinical course?

Response 2: The two patients who had PA bands as their first palliation both had their defect repaired < 24 months (16 months and 10 months). Both had multivessel PVS, but did not have significantly worse or better outcomes (patient 1: 1 surgery, 3 caths, diagnosed with PHTN; patient 2: 0 surgeries, 0 caths and no PHTN).